# Fluid Balance, Sweat Na^+^ Losses, and Carbohydrate Intake of Elite Male Soccer Players in Response to Low and High Training Intensities in Cool and Hot Environments

**DOI:** 10.3390/nu13020401

**Published:** 2021-01-27

**Authors:** Ian Rollo, Rebecca K. Randell, Lindsay Baker, Javier Yanguas Leyes, Daniel Medina Leal, Antonia Lizarraga, Jordi Mesalles, Asker E. Jeukendrup, Lewis J. James, James M. Carter

**Affiliations:** 1Gatorade Sports Science Institute, PepsiCo Life Sciences, Global R&D, Leicestershire LE4 1ET, UK; rebecca.randell@pepsico.com (R.K.R.); lindsay.baker@pepsico.com (L.B.); james.carter@pepsico.com (J.M.C.); 2School of Sports Exercise and Health Sciences, Loughborough University, Leicestershire LE11 3TU, UK; a.e.jeukendrup@lboro.ac.uk (A.E.J.); L.James@lboro.ac.uk (L.J.J.); 3FC Barcelona Medical Department, FC, 08014 Barcelona, Spain; xavier.yanguas@fcbarcelona.cat (J.Y.L.); damele@me.com (D.M.L.); mlizarraga@ub.edu (A.L.); j.Mesalles@fcbarcelona.cat (J.M.)

**Keywords:** hydration, fluid, carbohydrate, professional, soccer

## Abstract

Hypohydration increases physiological strain and reduces physical and technical soccer performance, but there are limited data on how fluid balance responses change between different types of sessions in professional players. This study investigated sweat and fluid/carbohydrate intake responses in elite male professional soccer players training at low and high intensities in cool and hot environments. Fluid/sodium (Na^+^) losses and ad-libitum carbohydrate/fluid intake of fourteen elite male soccer players were measured on four occasions: cool (wet bulb globe temperature (WBGT): 15 ± 7 °C, 66 ± 6% relative humidity (RH)) low intensity (rating of perceived exertion (RPE) 2–4, m·min^−1^ 40–46) (CL); cool high intensity (RPE 6–8, m·min^−1^ 82–86) (CH); hot (29 ± 1 °C, 52 ± 7% RH) low intensity (HL); hot high intensity (HH). Exercise involved 65 ± 5 min of soccer-specific training. Before and after exercise, players were weighed in minimal clothing. During training, players had ad libitum access to carbohydrate beverages and water. Sweat [Na^+^] (mmol·L^−1^), which was measured by absorbent patches positioned on the thigh, was no different between conditions, CL: 35 ± 9, CH: 38 ± 8, HL: 34 ± 70.17, HH: 38 ± 8 (*p* = 0.475). Exercise intensity and environmental condition significantly influenced sweat rates (L·h^−1^), CL: 0.55 ± 0.20, CH: 0.98 ± 0.21, HL: 0.81 ± 0.17, HH: 1.43 ± 0.23 (*p* =0.001), and percentage dehydration (*p* < 0.001). Fluid intake was significantly associated with sweat rate (*p* = 0.019), with no players experiencing hypohydration > 2% of pre-exercise body mass. Carbohydrate intake varied between players (range 0–38 g·h^−1^), with no difference between conditions. These descriptive data gathered on elite professional players highlight the variation in the hydration status, sweat rate, sweat Na^+^ losses, and carbohydrate intake in response to training in cool and hot environments and at low and high exercise intensities.

## 1. Introduction

The consequences of high-intensity intermittent running include an elevation in core temperature (39–40 °C) [1] and a gradual depletion of endogenous carbohydrates [2]. The subsequent thermoregulatory responses include an increased skin blood flow and the onset of sweating to allow evaporative heat loss [3]. A depletion of muscle glycogen and elevations in core temperature during exercise can be associated with fatigue, which, in soccer, may manifest as a reduction in overall distance covered and reduction in high-intensity running [4,5].

Despite fluid availability, soccer players have been reported to experience net fluid deficits during exercise because of fluid lost through sweat. Hypohydration has been reported to increase physiological strain [3] and is associated with reduced physical [6] and technical soccer performance [7], although not all studies report this [8]. Fluid losses vary greatly among elite soccer players, even in response to the same exercise conditions [9]. Furthermore, fluid losses are highly influenced by the environment [10,11] and exercise intensity [12]. Elite players’ sweat rates have been reported to be lower (1.13 ± 0.30 L·h^−1^, 0.71–1.77 L·h^−1^) when training in cool temperatures (5 ± 1 °C), in comparison with hot conditions (37 ± 3 °C) (1.46 ± 0.24 L·h^−1^, 1.12–2.09 L·h^−1^) [11]. Nevertheless, levels of hypohydration are still evident when training in cool environments (1.62 ± 0.55%, range 0.87%–2.55%) [10]. This is because training intensity will continue to help determine the level of metabolic heat production and corresponding sweat response [13].

Studies investigating fluid balance in professional soccer players have reported voluntary fluid intake to be highly varied among players on the same team and influenced by the environmental conditions. Maughan et al. (2005) tabulated results from five studies that investigated fluid loss and intake of elite male soccer players at temperatures ranging from 5 to 32 °C. Ad-libitum fluid intake ranged from 0.42 to 1.40 L [10].

Less information is available on carbohydrate intake during soccer training and matches. This is of interest because the routine ingestion of carbohydrates provided in combination with fluid, typically via carbohydrate–electrolyte beverages, has been reported to delay fatigue and positively influence various aspects of soccer-specific performance [14]. In addition, practicing carbohydrate intake during exercise of similar “match” intensities may help the player better tolerate carbohydrate ingestion during competition [15]. In a study assessing the overall energy intake and energy expenditure of professional soccer players, carbohydrate intake during training was reported to be significantly lower (3.1 ± 4.4 g·h^−1^) in comparison with a competitive match (32.3 ± 21.9 g·h^−1^) [16]. Although carbohydrate intake during training was collected on multiple days, the training intensity was consistently low (< 48 m·min^−1^). Therefore, to date, it has not been reported if professional players adjust their carbohydrate intake to an increase in training intensity.

Finally, post exercise, the ingestion of fluid and replenishment of electrolytes is required for rapid rehydration [17]. Sweat [Na^+^] has been reported in separate groups of professional players in both cool [10] and hot conditions [11] and at different intensities [12]. However, since these earlier studies, performance parameters in professional soccer have become more intense [18] and technologies have been developed to rapidly quantify sweat [Na^+^] [19].

Thus, the aim of the present study was to investigate the sweat response, ab-libitum fluid, and carbohydrate intake of the same group of elite male professional soccer players during training at low and high exercise intensities performed in cool and hot environments.

## 2. Materials and Methods

### 2.1. Study Participants

Thirty elite professional soccer players participated in the present study. Five players did not return to the club following the first test and 11 players were unavailable on two of the four test days. Therefore, 14 male first-team players of Futbol Club (FC) Barcelona (Spanish first division; La Liga) completed this study. All players completed a health screening questionnaire and provided written informed consent to participate after the details of the study had been explained. The study was approved by the Research Ethics Committee of Loughborough University, U.K. (R16-P133). All players were professional, accustomed to training and/or match durations of between 60 and 120 min 3–6 times per week. The physical characteristics of the players were age: 24 ± 4 years; body mass: 75.2 ± 6.2 kg; stature 180.9 ± 7.1 cm; V̇O_2max_: 57.9 ± 3.8 mL·kg BM·min^−1^ (V̇O_2max_ assessments were performed in July).

### 2.2. Experimental Design

The measurements were made on four separate training days during the competitive season. The study design was observational and thus the results are descriptive only of those training days on which the tests were completed. Training began at approximately 11.00 a.m. and lasted 65 ± 5 min. Each training session was completed outside on a grass pitch in dry conditions. Environmental temperature and relative humidity were recorded (Kesteral 4500, Nielsen-Kellerman, Pennsylvania, USA) at the beginning and at 15-min intervals during training. The training was that which was normally conducted by the players at that time of the season, with no interference from the research team. Furthermore, players were free-living with no research-requested dietary controls.

All players participated in the same training sessions. The training intensity was classified by a global positioning satellite system (STATSport, 10 Hz Viper [20]). In addition, at the cessation of training, players were asked to rate “how hard” the training session was on a 1–10 rating of perceived exertion (RPE) scale [21]. For the purpose of the present study, ratings of 1–5 were indicative of low intensity and ratings of 6–10 were indicative of high intensity (Table 1).

Data were collected under the following conditions classified by wet bulb globe temperature (WBGT): cool (14 ± 7 °C, 67 ± 7% relative humidity (RH)) low intensity (CL); cool (14 ± 8 °C, 69 ± 7 RH) high intensity (CH); hot (28 ± 1 °C, 55 ± 9 RH) low intensity (HL); hot (28 ± 2 °C, 55 ± 10 RH) high intensity (HH) [22]. Testing in the two hot conditions was completed in Barcelona, Spain (July). Testing in the cool high intensity condition was completed in the United Kingdom (July) and the cool low intensity condition was completed in Barcelona, Spain (January) of the same competitive season. The testing schedule was dictated by players’ availability during the pre-season and winter break schedule. Players self-selected the clothing worn on each testing occasion. Outfield players wore a training shirt, shorts, soccer socks, and soccer boots in hot conditions. In the cool conditions, players also wore base layers that covered limbs and torso, and neck scarves. During the training sessions, all players had free access to water and a 6% carbohydrate–electrolyte beverage (Gatorade Thirst Quencher, PepsiCo Ltd.) in individually labelled drinks bottles, but drinking was limited to during coach-allocated “drinks breaks”.

### 2.3. Assessment of Fluid Balance

A pre-exercise urine sample was collected from all players 30–60 min prior to exercise. These samples were analysed for urine specific gravity (USG) (Atago 3730 Pen-Pro Dip-Style Digital Refractometer, Washington, USA) to provide an indication of pre-training hydration status. Hydration status was classified as follows: < 1.020 euhydrated, 1.021–1.024 minimally hypohydrated, > 1.024 hypohydrated [23]. Containers were allocated to each player to collect any urine passed during practice. In the hour before training, players were encouraged to void their bladder before body mass (kg) was recorded in minimal clothing. Sweat loss was calculated from the change in body mass collected after exercise, following the correction for fluid intake and any urine/stool loss [24]. Relatively small changes in mass due to substrate oxidation and other sources of water loss were ignored [25]. 

### 2.4. Assessment of Sweat [Na^+^]

Full details of the sweat collection protocol have been described in detail by Baker and colleagues (2014). In summary, sweat patches (Tegaderm + Pad, 3M, Loughborough, UK) were applied to specific landmarks on the surface of the thigh and back (in case no sample was available from the thigh). In all 14 players across all four conditions, the thigh patch was used with no exceptions. The sweat samples were analysed with a compact wireless analyser that uses ion-selective electrode technology to derive measures of [Na^+^] (Horiba B-722) [19]. Regional sweat [Na^+^] was normalised to whole-body concentrations [26]. Sodium chloride losses per hour of exercise were calculated from the sweat [Na^+^] and individual sweat rate [24].

### 2.5. Statistical Analysis

Data are reported as mean and standard deviation (mean, s), with the range of data given in parentheses. All data were analysed using Minitab software package (version 17; Minitab Inc, State College, PA, USA). A Shapiro–Wilk test was used to determine normality of distribution. All variables (sweat rate, percentage dehydration, sweat [Na^+^], sodium chloride loss, fluid intake, and carbohydrate intake) were checked for homoscedasticity using the Levene test, and analysed using a one-way repeated measures analysis of variance (ANOVA), and with Tukey’s Honestly Significant Difference (HSD) pairwise comparison when a significant main effect was identified. A Pearson Correlation Coefficient was used to determine the strength of the relationship between USG and sweat rate, and sweat rate and fluid intake. Statistical significance was declared when *p* < 0.05.

## 3. Results

The players’ physical performance during the training sessions was classified as low or high-intensity exercise by the parameters listed in Table 1. Pre-exercise body mass was not different between trials 75.2 ± 6.2 kg (67.1–86.0 kg) (*p* > 0.05). Three of the 14 players (21%) (four players (29%) in the HL trial) had urine specific gravity indicative of “euhydration” prior to each trial. The majority of players, 57%, 64%, 57%, and 50%, had urine specific gravity indicative of “hypohydration” prior to the CL, CH, HL, and HH trials, respectively.

The fluid balance variables for the four exercise conditions were normally distributed (Shapiro–Wilk test, *p* = 0.328) and are reported in Table 2. Levene tests were run on each dataset, comparing all four levels to check for equality of variance. As *p* > 0.05 for every test, we can assume that variances are equal across groups. No urine was produced by players during the training sessions. For each trial, there were three drinks breaks per hour of exercise. The duration of each drinks break was 90 ± 30 s. The mean and range of carbohydrate intake are reported in Table 2. The frequency of players ingesting 0 g of carbohydrate was *n* = 2 for the CL trial, *n* = 4 in the CH trial, *n* = 1 for the HL trial, and *n* = 2 in the HH trial. Three of the 14 players showed consistency in their choice of beverage, opting for no carbohydrate in two to three of the four trials. Nine of the 14 players ingested carbohydrates during all exercise conditions.

There was no association between pre-exercise urine specific gravity and sweat rate (*r* = 0.065, *r*^2^ = 0.004, *p* > 0.05). The relationship between sweat rate (L·h^−1^) and the volume of drink consumed (mL·h^−1^) during all training sessions was significant (*p* = 0.019) (Figure 1). The respective sodium chloride losses during exercise and carbohydrate intake are reported in Table 2. The distribution and central tendency of player percentage dehydration are displayed in Figure 2 and player sweat [Na^+^] is displayed in Figure 3. Mean sweat [Na^+^] did not change significantly over the four testing occasions. The individual change in sweat [Na^+^] over the four trials was 10 ± 10 mmol·L^−1^ (3.6–16.7 mmol·L^−1^).

## 4. Discussion

The aim of the present study was to investigate the fluid balance and carbohydrate intake of elite professional male soccer players in response to exercise at different intensities and in different environmental conditions. Consistent with the literature in this population, all players experienced a reduction in body mass over the duration of the training sessions. However, the novel findings of the present study were: (1) players adequately adjusted ad-libitum fluid intake to prevent hypohydration greater than 2% of pre-exercise body mass; (2) carbohydrate ingestion was no different between conditions, with a large range in the rate of ingestion between individual players; and (3) thigh sweat [Na^+^] was not different across conditions.

In the present study, sweat rates were similar to those reported in the soccer literature (1.0–2.2 L·h^−1^) [9,28]. Furthermore, the change in sweat rate across conditions was similar to those previously reported in hot (~35 °C) (1.48 ± 0.36 L·h^−1^) [11] and cool environments (5 °C) (1.13 ± 0.30 L·h^−1^) [10] and in response to an increase in exercise intensity [12]. However, the present study is the first to show this pattern of responses in the same group of elite professional players in the same competitive season. Interestingly, a player’s level of hypohydration during the cool high intensity condition was not significantly different from high intensity exercise in the heat. Our observations would suggest that, beyond exercise intensity, this may be a consequence of players wearing more clothing in response to the cold environment. Hypohydration ensues when multiple layers of clothing increase sweat rates [29], combined with reduced fluid intake when exercising in cool environments [10].

In contrast to previous studies, no player experienced hypohydration > 2% of pre-exercise body mass values [9,12]. This observation may be due to the shorter duration of exercise in the present study (~60 min), to that reported previously (100 min) [9]. Fluid loss is of interest in professional soccer because acute hypohydration has been reported to lead to decrements in both physical and technical performance [7]. The typical threshold for hypohydration to impact exercise performance is reported as 2% of pre-exercise values [30]. However, this assumes a pre-exercise hydration status of “euhydration”. The urine specific gravity data suggest that this was not always the case. In professional soccer, pre-exercise hypohydration as indicated by urine concentration is common [25,31]. Consequently, it is likely that the resulting fluid balance data underestimate the extent of subsequent hypohydration that occurs during exercise. Interestingly, recent evidence suggests that routine exposure to hypohydration, potentially such as that resulting from day-to-day training in soccer, may alleviate some performance detriments [32]. Whilst the limitations of urine specific gravity as a “hydration assessment” are recognised, more invasive measures, such as measuring plasma osmolality, were not possible in the present study [33]. Despite potential “familiarisation” to hypohydration, it is important to note that performance remains “optimised” by commencing exercise euhydrated [32]. Thus, care should be taken that players are not simply accustomed to hypohydration. To achieve this, appropriate rehydration strategies are recommended between exercise sessions [17]. In addition, daily changes in body mass, thirst, and urine USG may be collected to track changes in players’ hydration status [34].

The mean carbohydrate intake was approximately 15 g·h^−1^ and remained unchanged between exercise occasions (Table 2). The guidelines for carbohydrate intake for exercise durations of 60–70 min, especially during training as in the present study, are a “grey area” in sports nutrition. Specifically, the most recent The American College of Sports Medicine( ACSM) position recommends “small amounts of carbohydrate including mouth rinse” for exercise durations of 45–75 min, and 30–60 g of carbohydrate per hour for durations of 1–2.5 h, “including stop and start” sports [23]. Therefore, it is reasonable to suggest that most players met the minimum recommendation (i.e., small amounts), especially as the players were fed, and that exercise durations were closer to 1 h than to 2.5 h. The requirement for carbohydrate ingestion during exercise depends upon the demands and duration of the exercise and the goal of the individual player [35]. Thus, during low-intensity sessions, in cool environments, there may be little benefit of carbohydrate ingestion. Furthermore, some aerobic adaptation may be augmented by abstaining from carbohydrate ingestion before and during exercise [36]. Conversely, carbohydrate ingestion is associated with improved physical and technical performance in soccer [37]. Thus, it is advised that players practice ingesting those volumes and quantities associated with improved performance [15]. Glycogen stores will be depleted during 60 min of soccer activity [5] and the rate of glycogen use is elevated when playing in hot environments [38]. Nevertheless, there is no evidence that increasing the quantity of carbohydrate ingested (to quantities recommended for exercise durations > 3 h, 90 g carbohydrate h^−1^) would result in further physical and technical performance benefits [23,39].

Previous studies have reported sweat [Na^+^] and subsequent sodium chloride losses in different squads of professional male players in response to single training sessions [10,11]. Duffield et al. (2012) reported that sweat [Na^+^] of the same squad of professional players increased in response to higher exercise intensities. This finding was consistent with laboratory-based investigations that report a proportional increase in forearm sweat [Na^+^] as sweat rate is increased [40]. This is because at higher sweat rates Na^+^ secretion increases proportionally faster than the rate at which Na^+^ can be reabsorbed along the distal duct of the sweat gland [35]. In the present study, only four of the 16 players’ sweat [Na^+^] changed sufficiently to re-classify their values from low (< 36 mmol·L^−1^) to moderate (36–58 mmol·L^−1^) [41]. One possible reason for the discrepancy between our results and previous studies could be differences in the site of sweat collection. The study by Duffield and colleagues collected sweat from the scapula, whereas in the present study it was collected from the thigh. In a recent laboratory study, sweat rate and sweat [Na^+^] significantly increased on several upper body regions (including the scapula and forearm) in response to increased exercise intensity, but there were no changes on the lower body (thigh and calf) [42]. Of course, this is difficult to confirm in the present study without concomitant local sweat rates on the thigh or data from additional anatomical regions. Future field research measuring sweat [Na^+^] from several upper and lower body sites is needed. Nevertheless, sodium chloride loss significantly increased in the heat and at higher intensities because of the increased sweat rate. Thus, the results of the present study support previous recommendations that re-hydration strategies be modified to the exercise intensity, environmental condition, and individual player [12,28].

### 4.1. Limitations, Strengths, and Future Research

The present study was descriptive and thus several limitations are acknowledged. First, the final assessment was completed 5 months after the first tests completed in July. It is unknown if the players’ acclimation and/or physical status changed over this duration, which may have influenced both the sweat rate and sweat [Na^+^] response to exercise. Second, the analysis of sweat did not include potassium concentration, which would have been preferential as a quality control for sweat samples [43]. Despite this, best practices of sweat collection (clean skin, avoided patch saturation, gloves, etc.) described by Baker [43] were followed, so that we have confidence in our analysis. Other factors reported to influence sweat [Na^+^] such as hormones [44] and diet [45] were not recorded or controlled for prior to testing, due to the nature of field/descriptive studies. Another limitation of the present study was that fluid intake was only assessed during exercise. The pre-training spot USG measure did not allow for the complete understanding of the players’ hydration status before training. Assessing first morning USG, 24-h urine volumes, and/or 24-h fluid intake prior to training would provide insight into the influence of baseline hydration on sweat rates and sweat [Na^+^]. Finally, as some (*n* = 3) players ingested no carbohydrate during exercise, future studies could investigate how alternative sources of carbohydrate (i.e., gels, bars, chews) may influence the ad-libitum carbohydrate intake of players. This would be relevant during high-intensity training sessions and prolonged pre-season training, and to encourage practicing match day nutrition.

### 4.2. Practical Applications

This study shows that observations such as the variability in sweat rate and sweat [Na^+^] made in sub-elite soccer populations also apply to professional players. This study is evidence that it is possible to gather hydration-related data on elite players without disruption to normal training or competitive schedules. Furthermore, the results of the present study suggest that sweat [Na^+^] does not change over time or between different sessions. Therefore, this kind of sweat [Na^+^] analysis, often the expensive and technically time-consuming part of the process, may be completed on a single occasion. The sweat [Na^+^] can then be applied to sweat rates, which are more easily measured on multiple occasions. However, further studies are required to confirm this applies over the entirety of the season.

## 5. Conclusions

These descriptive data gathered on elite professional players highlight the variation in the hydration status, sweat rate, sweat Na^+^ loss, and carbohydrate intake in response to training in cool and hot environments and at low and high exercise intensities. Thus, these data support recommendations that fluid and carbohydrate intake strategies should be specific to the individual and exercise occasion.

## Figures and Tables

**Figure 1 nutrients-13-00401-f001:**
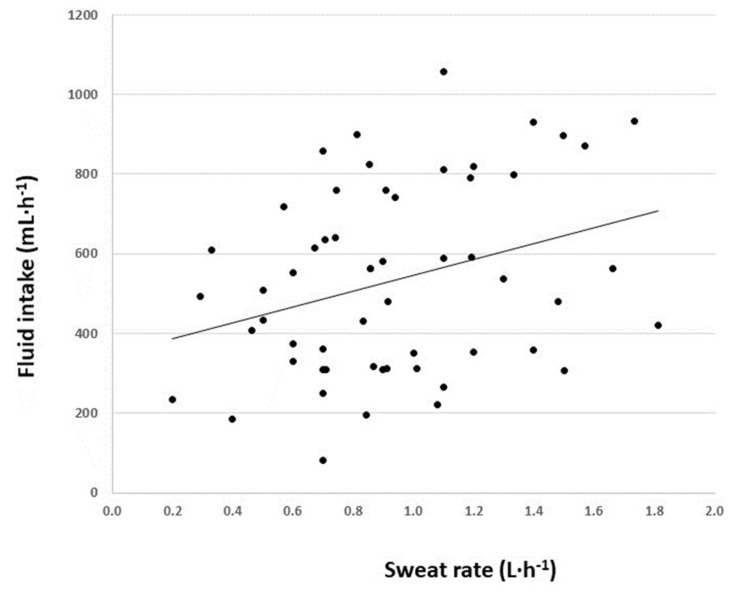
The relationship between sweat rate (L·h^−1^) and the volume of drink consumed (mL·h^−1^) during all training sessions was significant (*p*= 0.019, *r* = 0.31, *r*^2^ = 0.098).

**Figure 2 nutrients-13-00401-f002:**
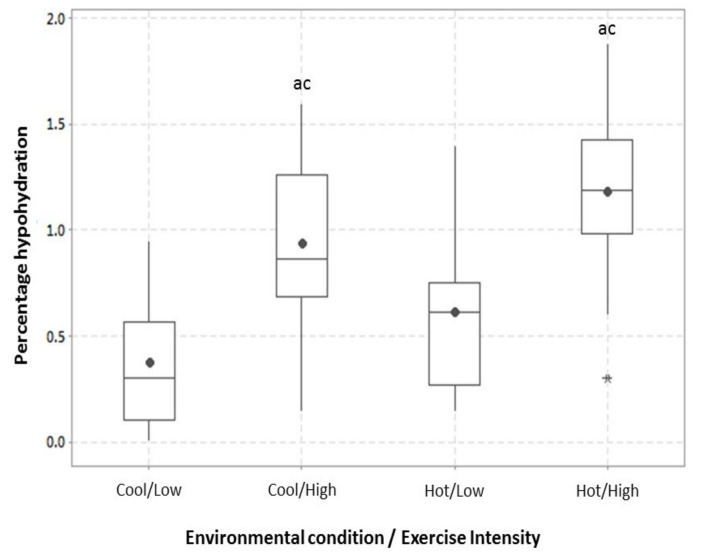
The percentage of hypohydration experienced by players in response to exercise at low and high exercise intensities in cool and hot conditions. The mean (●), median (-), interquartile range box (middle 50% of the data), and upper and lower error bars representing the upper and lower 25% of the distribution, respectively, are displayed. a = significantly different from Cool Low, c = significantly different from Hot Low. * indicates outlier in results.

**Figure 3 nutrients-13-00401-f003:**
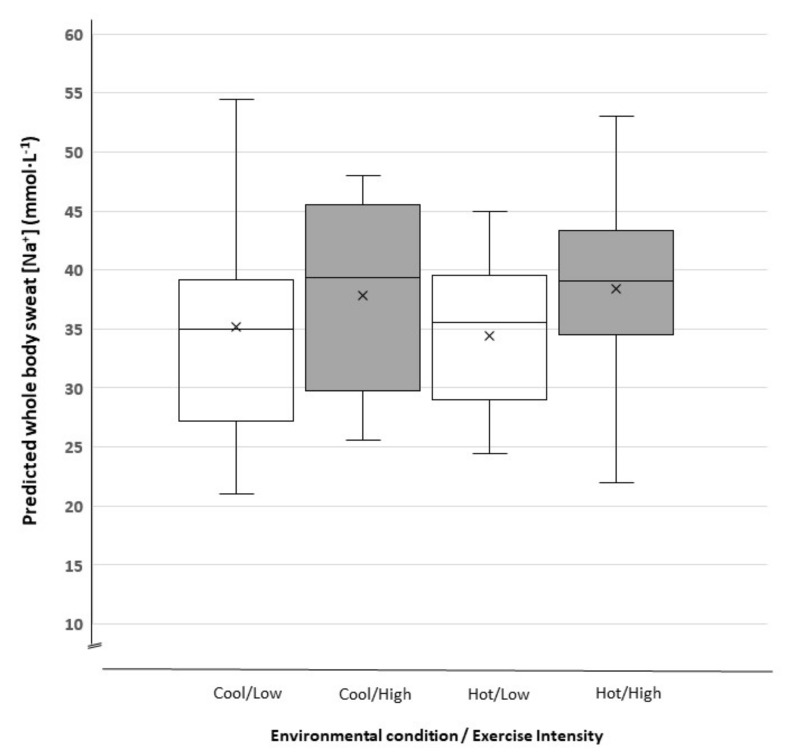
Predicted whole-body sweat sodium concentration (mmol·L^−1^; from thigh sweat [Na^+^]) during exercise at low and high exercise intensities in cool and hot conditions. The mean (x), median (-), interquartile range box (middle 50% of the data), and upper and lower error bars representing the range. No significant difference between trials (*p* > 0.05).

**Table 1 nutrients-13-00401-t001:** Global positioning satellite parameters and player rating of perceived exertion (RPE) for training intensity classification. High speed running defined as > 14 km·h^−1^ < 20 km·h^−1^. Sprint defined as > 20 km·h^−1^.

Parameter	Intensity	
	Low	High
Rating of Perceived Exertion	2–4	6–8
Total distance (m)	2509–2593	4889–4949
m·min^−1^	40–46	82–86
High Speed Running (m)	33–35	191–232
Number of Sprints	2–3	10–12

**Table 2 nutrients-13-00401-t002:** The mean ± *s* and (range) of variables recorded before and during each training session in response to exercise at low and high exercise intensities in cool and hot conditions. Predicted whole-body sweat [Na^+^], predicted whole-body sweat Na^+^ losses, and assumed sodium chloride losses during exercise. The loss of sodium chloride is calculated on the assumption that all Na^+^ loss is sodium chloride [27]. a = significantly different from Cool Low, b = significantly different from Cool High, c = significantly different from Hot Low.

	Condition					
	Cool		Hot			
Intensity	Low	High	Low	High	*F*	*p*
Urine Specific Gravity	1.024 ± 0.005	1.023 ± 0.004	1.023 ± 0.006	1.025 ± 0.005	(3,52) = 0.35	0.786
	(1.016–1.032)	(1.014–1.028)	(1.011–1.033)	(1.017–1.034)		
Sweat rate (L·h^−1^)	0.55 ± 0.20	0.98 ± 0.21 ^a^	0.81 ± 0.17 ^a^	1.43 ± 0.23 ^abc^	(3,52) = 46.37	0.001
	(0.20–0.85)	(0.67–1.50)	(0.50–1.20)	(1.10–1.81)		
Fluid intake (mL·h^−1^)	394 ±160	505 ± 265	572 ± 214	663 ± 229 ^a^	(3,52) = 3.68	0.018
	(184–719)	(220–1058)	(308–898)	(266–933)		
Carbohydrate intake (g·h^−1^)	12 ± 9	11 ± 11	15 ± 12	15 ± 14	(3,52) = 0.57	0.637
	(0–37)	(0–32)	(0–36)	(0–50)		
Sweat [Na^+^] (mmol·L^−1^)	35 ± 9	38 ± 8	34 ± 7	38 ± 8	(3,52) = 0.85	0.475
	(21–54)	(26–48)	(24–45)	(22–53)		
Sweat Na^+^ loss (mmol·h^−1^)	19 ± 9	38 ± 13 ^a^	28 ± 9 ^a^	54 ± 15 ^abc^	(3,52) = 22.98	0.001
	(7–36)	(19–68)	(13–52)	(33–81)		
NaCl loss (g·h^−1^)	1.1 ± 0.5	2.2 ± 0.8 ^a^	1.6 ± 0.5 ^a^	3.2 ± 0.9 ^abc^	(3,52) = 22.98	0.001
	(0.4–2.1)	(1.1–3.9)	(0.8–3.0)	(1.9–4.8)		

## Data Availability

The data presented in this study are available on request from the corresponding author and the permission of all parties involved in the study. The data are not publicly available due to privacy.

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
