# Peer review of "Fluid Balance, Sweat Na+ Losses, and Carbohydrate Intake of Elite Male Soccer Players in Response to Low and High Training Intensities in Cool and Hot Environments"

_nutrients, 2021, doi:10.3390/nu13020401_

Round 1

Reviewer 1 Report

Although this study does not present very new results, the population (elite soccer players) and the methodology used give it an important value. However, there are some points that can be improved.

  1. Please separate the materials and methods by section.
  2. Please improve the wording of the result section. Also in the correlation graphs add the value r2 or r. For example, in graph 1 the value of r2 does not appear, neither in the text nor in the graph.
  3. Please unify the graphics (for example the colors), for example graphics 2 and 3 are very different (shape), even though they have the same groups.
  4. Please explain what you mean by "thigh corrected sweat [Na +] during exercise at low and high exercise intensities in cool and hot conditions".
  5. Please in table 1 add the p-values ​​of the comparisons.
  6. Please explain "The loss of sodium chloride is calculated on the basis that all the Na + loss is sodium chloride". Was sodium chloride measured?
  7. The conclusions are confusing and do not agree with the results. For example, "sweat [Na +] were not different across conditions", but in Table 2, there are differences between "Sweat Na + loss (mmol · h-1)". Please be more specific.
  8. The discussion is very poor, please; -Add a paragraph of study limitations, for example not measuring potassium which is used as a quality control for sweat samples. "Baker, L.B. (2017). Sweating rate and sweat sodium concentration in athletes: A review of methodology and intra / interindividual variability. Sports Medicine, 47 (Suppl. 1), 111–128." -
  9. Add a discussion paragraph to analyze potential variables that influence the loss of sodium and fluids in different conditions (climate / intensity). ... Castro-Sepulveda M, Cancino J, Fernández-Verdejo R, Pérez-Luco C, Jannas-Vela S, Ramirez-Campillo R, Del Coso J, Zbinden-Foncea H. Basal Serum Cortisol and Testosterone / Cortisol Ratio Are Related to Rate of Na + Lost During Exercise in Elite Soccer Players. Int J Sport Nutr Exerc Metab. 2019 Oct 17: 1-6: McCubbin AJ, Lopez MB, Cox GR, Caldwell Odgers JN, Costa RJS. Impact of 3-day high and low dietary sodium intake on sodium status in response to exertional-heat stress: a double-blind randomized control trial. Eur J Appl Physiol. 2019 Sep; 119 (9): 2105-2118.

Author Response

Manuscript ID

nutrients-1052790

Reviewer 1

General Comment

Although this study does not present very new results, the population (elite soccer players) and the methodology used give it an important value. However, there are some points that can be improved.

General response

Thank you for the review and we appreciate the constructive comments. We have taken all comments with positive intent and as such believe the responses have made considerable improvements to the manuscript. We have included the response to the specific comments below, detailing the modifications and marking the changes to the manuscript in blue text. 

Specific comments

Comment 1: Please separate the materials and methods by section.

Response: page 3-4: The material and methods has been separated by section, 2.1-2.5

Comment 2: Please improve the wording of the result section. Also in the correlation graphs add the value r2 or r. For example, in graph 1 the value of r2 does not appear, neither in the text nor in the graph.

Response: In line with comments from other reviewers the sections of results section have been rewritten. The r2 value has been added to the legend.

Comment 3 :Please unify the graphics (for example the colors), for example graphics 2 and 3 are very different (shape), even though they have the same groups.

Response: The figures 2 and 3 have been reformatted to be more uniform in shape and size.

Comment 4: Please explain what you mean by "thigh corrected sweat [Na +] during exercise at low and high exercise intensities in cool and hot conditions".

Response: The figure legend has been updated to explain that the “thigh corrected” values are “predicted whole body sweat sodium concentration”. The figure axis has been updated and legend now reads as follows:

Figure 3. Predicted whole body sweat sodium concentration (mmol·l-1; from thigh sweat [Na+]) during exercise at low and high exercise intensities in cool and hot conditions.

Comment 5: Please in table 1 add the p-values ​​of the comparisons.

Response: Thank you. Table 1 simply lists the parameters which were used to differentiate between the exercise intensities. i.e. high and low. They are descriptive and by definition substantially different. We are do not feel adding a statistical comparison between the criteria would be a meaningful add to the manuscript. 

Comment 6: Please explain "The loss of sodium chloride is calculated on the basis that all the Na + loss is sodium chloride". Was sodium chloride measured?

Response: Thank you. Instead of “basis” we have modified the sentence to state “assumption” and cited an appropriate reference (Baker et al. JAP 2018) which shows an almost 1:1 ratio of whole body sodium and chloride concentration (on a mmol/L basis). 

The loss of sodium chloride is calculated on the assumption that

Comment 7:The conclusions are confusing and do not agree with the results. For example, "sweat [Na +] were not different across conditions", but in Table 2, there are differences between "Sweat Na + loss (mmol · h-1)". Please be more specific.

Response: To clarify, we are reporting two separate variables. Sweat sodium concentration and total sweat sodium loss (which will be a product of sweat concentration and the volume of sweat lost i.e. sweat rate).

To be more specific in the text we have changed in the legend to clarify. The legend now reads:

Table 2. The mean ± s and (range) of variables recorded before and during each training session in response to exercise at low and high exercise intensities in cool and hot conditions. Predicted whole body sweat [Na+], predicted whole body sweat Na+ losses, and assumed sodium chloride losses during exercise. The loss of sodium chloride is calculated on the assumption that all the Na+ loss is sodium chloride [3]. a = significantly different from Cool Low, b = significantly different from Cool High, c = significantly different from Hot Low.

Comment 8: The discussion is very poor, please; -Add a paragraph of study limitations, for example not measuring potassium which is used as a quality control for sweat samples. "Baker, L.B. (2017). Sweating rate and sweat sodium concentration in athletes: A review of methodology and intra / interindividual variability. Sports Medicine, 47 (Suppl. 1), 111–128." –

Response:

We acknowledge that the discussion on version 1 was limited. Therefore in line with the request from other reviewers we have added two additional sections to the manuscript discussion, specific to limitations as well as on practical implications. Both sections are added to text and read as follows:

Limitations, strengths, and future research

The present study was descriptive and thus several limitations are acknowledged. First, the final assessment was completed 5 months after the first tests completed in July. It is unknown if the players acclimation and/or physical status changed over this duration, which may have influenced both the sweat rate and sweat [Na+] response to exercise. Second, the analysis of sweat did not include potassium concentration which would have been preferential as a quality control for sweat samples [43]. Despite this, best practice of sweat collection (clean skin, avoided patch saturation, gloves, etc.) described by Baker [43] were followed, so that we have confidence in our analysis.  Other factors reported to influence sweat [Na+] such as hormones [44] and diet [45] were not recorded or controlled for prior to testing, due to the nature of field/descriptive studies. Finally, as some (n=3) players ingested no carbohydrate during exercise, future studies could investigate how alternative sources of carbohydrate (i.e. gels, bars, chews) may influence ad-libitum carbohydrate intake of players. This would be relevant during high intensity training sessions, prolonged pre-season training and to encourage practicing match day nutrition.

Practical applications

This study shows that observations, such as the variability in sweat rate and sweat [Na+] observed in sub-elite soccer populations also apply to professional players. This study is evidence that it is possible to gather hydration related data on elite players without disruption to normal training or competitive schedules. Furthermore, the results of the present study suggest sweat [Na+] do not change over time or between different sessions. Therefore, this sweat [Na+] analysis, often the expensive and technically time-consuming part of the process, may be completed on a single occasion. The sweat [Na+] can then be applied to sweat rates, which are more easily measured on multiple occasions. However, further studies are required to confirm this applies over the entirety of the season.

Comment 9: Add a discussion paragraph to analyze potential variables that influence the loss of sodium and fluids in different conditions (climate / intensity). ... Castro-Sepulveda M, Cancino J, Fernández-Verdejo R, Pérez-Luco C, Jannas-Vela S, Ramirez-Campillo R, Del Coso J, Zbinden-Foncea H. Basal Serum Cortisol and Testosterone / Cortisol Ratio Are Related to Rate of Na + Lost During Exercise in Elite Soccer Players. Int J Sport Nutr Exerc Metab. 2019 Oct 17: 1-6: McCubbin AJ, Lopez MB, Cox GR, Caldwell Odgers JN, Costa RJS. Impact of 3-day high and low dietary sodium intake on sodium status in response to exertional-heat stress: a double-blind randomized control trial. Eur J Appl Physiol. 2019 Sep; 119 (9): 2105-2118.

Response:

Thank you for directing us to these references. We have included in the discussion under the study limitation section. Please see previous response.

Reviewer 2 Report

Keywords: hydration 1, fluid 2, carbohydrate 3, professional 4, soccer 5. Delete number after

CARBOHYDRATES: Use CHO through the text

Abstract

Include brief background and objective. Rewrite conclusions

Studies investigating fluid balance in professional player: Include Professional soccer players  

Highly varied between players: Include Among players 

Materials and Methods

Research Ethics Committee of Loughborough University, UK. INCLUDE ETHICAL ISSUES NUMBER

V̇O2max : 57.9 ± 3.8 ml·kg BM·min-1. 2 MAX IN SUBINDEX

Global positioning satellite system (GPS; list make and model). Describe the Wimu system. Reference and validity 

For the purpose of the present study ratings of 1-5 were classified as low intensity and ratings of 6-10 were classified as high intensity. Describe this criterion

Sodium chloride losses per hour of exercise were calculated from the sweat [Na+ ] and individual sweat rate. Include reference 

Any urine passed during training. Include during practice

Sweat loss was calculated from the change in body mass collected after exercise following the correction for fluid intake and any urine/stool loss. Include reference 

Testing in the cool conditions was completed in the UK July 2014 and in Spain January 2015. Explain with details this sentence 

A Kolmogorov-Smirnov test ...? initial n=30, better SHAPIRO WILK 

Did you check the homoscedasticity of the data by LEVENE test?

Considering that the intensity could vary the level of sweating, perhaps this parameter should be included as a covariate in the statistics

RESULTS

There was no association between the pre-exercise urine specific gravity and sweat rate or fluid intake in ml·h -1 (r2 = 0.002, r2 =0.003, P>0.05, respectively). Please include r and r2 in statistical part

Figure 1. Include r and p values

Table 2. Words and numbers in smaller size  

sweat [Na+]: + in superindex. Include through the text  

as some (n.=3) players. Delete point after n

DISCUSSION

Carbohydrates section: Taking into account that some authors suggest that carbohydrates intake during long time (>3 h) exercise could be higher than ACSM recommendations (120 g/h vs. 90 g/h), could this same discrepancy exist in the recommendations for shorter time activities? After all, energy systems for high intensity actions such as those performed in soccer require a high amount of glycogen and temperature variations can cause these requirements to increase. Discuss this.

Include limitations, strengths, and future research lines section

Include practical applications section

Author Response

Reviewer 2

General Comment

General response

Thank you for the review. We have taken all comments with positive intent and as such believe the responses have made improvements to the manuscript. We have included the response to the specific comments below, detailing the modifications and marking the changes to the manuscript in blue text. 

Comments

Abstract

 Comment 1: Include brief background and objective. Rewrite conclusions

 Response:

The abstract has been modified to include brief background, objective and conclusions have been rewritten.

Comment 2: Studies investigating fluid balance in professional player: Include Professional soccer players  

Response: This sentence has been re-written to include “professional soccer players”

Comment 3:Highly varied between players: Include Among players 

 Response:  This sentence has been rewritten: “be highly varied among players in…”

Materials and Methods

Comment 4 :Research Ethics Committee of Loughborough University, UK. INCLUDE ETHICAL ISSUES NUMBER

Response:  The ethical issue number has been added to the manuscript: R16-P133

Comment 5: V̇O2max : 57.9 ± 3.8 ml·kg BM·min-12 MAX IN SUBINDEX

Response: the 2 max has been formatted in sub index.

Comment 6: Global positioning satellite system (GPS; list make and model). Describe the GPS system. Reference and validity 

Response: the GPS make and model have been included as well as supporting reference for the validity.

(STATSport, 10 Hz Viper [1])

 Comment 7: For the purpose of the present study ratings of 1-5 were classified as low intensity and ratings of 6-10 were classified as high intensity. Describe this criterion

 Response: Further details have been added to this section to describe the criterion.

Comment 8: Sodium chloride losses per hour of exercise were calculated from the sweat [Na+ ] and individual sweat rate. Include reference 

Response: A reference has been added to support the sentence:

  1. Barnes, K.A., et al., Normative data for sweating rate, sweat sodium concentration, and sweat sodium loss in athletes: An update and analysis by sport. J Sports Sci, 2019. 37(20): p. 2356-2366.

Comment 9: Any urine passed during training. Include during practice

 Response: The sentence has been changed to during practice”

Comment 10: Sweat loss was calculated from the change in body mass collected after exercise following the correction for fluid intake and any urine/stool loss. Include reference 

Response: An appropriate reference has been added.

Comment 11: Testing in the cool conditions was completed in the UK in July and in Spain in January. Explain with details this sentence 

Response: A sentence has been added:

Testing schedule was dictated by players availability during the pre-season and winter break schedule.

Comment 12: A Kolmogorov-Smirnov test ...? initial n=30, better SHAPIRO WILK 

Response:

We changed the test for normality from the Kolmogorov-Smirnov to the Shapiro Wilk test as suggested. It did not influence the normality of data, and subsequent statistics. We acknowledge it is better to use Shapiro-Wilks test when sample size is <50.

(normally distributed (Shaprio-Wilks test, P=0.328)

The statistics section has been updated.

Comment 13: Did you check the homoscedasticity of the data by LEVENE test?

Response. Yes, however, our oversite was not report in the statistical analysis section or in the results. Therefore the statistical analysis section and results sections have been updated. The Levene test allowed us to check that variances are equal for our sample variables, and that any minor differences are down to random sampling. This is a necessary assumption for completing tests such as ANOVA. Levene tests were run on each dataset, comparing all 4 levels to check for equality of variance. As p>0.05 for every test, we can assume that variances are equal across groups.

Comment 14: Considering that the intensity could vary the level of sweating, perhaps this parameter should be included as a covariate in the statistics

Response: Thank you, this would certainly be interesting to explore and we will incorporate this into our future work were we can more tightly control conditions (i.e. lab-based testing), to give a better insight into mechanisms. We do not feel this is warranted in the present study that was designed to be a descriptive study to help inform practice in sports nutritionists/dieticians. That said, it seems unlikely that this would change the interpretation since the sweat rate changes of players were large and consistent between players, but the sweat [Na+] data changed little. For example, if we compare the low-cool condition with the high-hot condition, we can see the mean sweat rate responses are almost 3 fold different (0.5 vs 1.4 L/h), whilst the mean [Na+] is only 1.09 fold different (35 vs 38 mmol/L). Therefore, whilst some previous laboratory-based research suggests sweat rate is an important driver of sweat [Na+], this study does not fully support that. More importantly, we do not want to detract from the strong applied message here and that is that sweat sodium losses do not meaningfully differ between sessions and that practitioners likely do not need to make different assessments of sweat [Na+] to inform their practice.

RESULTS

Comment 15: There was no association between the pre-exercise urine specific gravity and sweat rate Please include r and r2 in statistical part

Response:  

The r and r2 value has been added to the statistical part.

Comment 16: Figure 1. Include r and p values

Response:  the P, r and r2 values have been added to the Figure 1 legend.

Comment 17: Table 2. Words and numbers in smaller size  

 Response: Thank you. The words and text have been reduced in size consistent with MS text. This has also been applied to figure and table legends.

Comment 18: sweat [Na+]: + in superindex. Include through the text  

 Response: the [Na+]: + has been superindexed throughout MS. [Na+

Comment 19: as some (n.=3) players. Delete point after n

 Response: the point has been deleted after the n. some (n=3)

DISCUSSION

Comment 20: Carbohydrates section: Taking into account that some authors suggest that carbohydrates intake during long time (>3 h) exercise could be higher than ACSM recommendations (120 g/h vs. 90 g/h), could this same discrepancy exist in the recommendations for shorter time activities? After all, energy systems for high intensity actions such as those performed in soccer require a high amount of glycogen and temperature variations can cause these requirements to increase. Discuss this.

 Response: Thank you. As stated the high quantities of carbohydrate ingestion are recommended for prolonged exercise >3 h in duration. Thus, this is not appropriate or relevant to soccer training or even competitive play (max duration 120 min). Nevertheless, there may be some confusion for the reader. Thus, we have included the section below and supporting references to clarify:

Glycogen stores will be depleted during 60 min of soccer activity [5] and the rate of glycogen use is elevated when playing in hot environments [6]. Nevertheless, there is no evidence that increasing the quantity of carbohydrate ingested (to quantities recommended for greater duration exercise >3 h, 90 g carbohydrate ·h-1) would result in further physical and technical performance benefits [7, 8].

Comment 21:Include limitations, strengths, and future research lines section

Response:

A limitations, strengths, and future research section has been added to the manuscript which now reads:

Limitations, strengths, and future research

The present study was descriptive and thus several limitations are acknowledged. First, the final assessment was completed 5 months after the first tests completed in July. It is unknown if the players acclimation and/or physical status changed over this duration, which may have influenced both the sweat rate and sweat [Na+] response to exercise. Second, the analysis of sweat did not include potassium concentration which would have been preferential as a quality control for sweat samples [43]. Despite this, best practice of sweat collection (clean skin, avoided patch saturation, gloves, etc.) described by Baker [43] were followed, so that we have confidence in our analysis.  Other factors reported to influence sweat [Na+] such as hormones [44] and diet [45] were not recorded or controlled for prior to testing, due to the nature of field/descriptive studies. Finally, as some (n=3) players ingested no carbohydrate during exercise, future studies could investigate how alternative sources of carbohydrate (i.e. gels, bars, chews) may influence ad-libitum carbohydrate intake of players. This would be relevant during high intensity training sessions, prolonged pre-season training and to encourage practicing match day nutrition.

Comment 22: Include practical applications section

A practical applications has been added to the manuscript which now reads:

Practical applications

This study shows that observations, such as the variability in sweat rate and sweat [Na+] observed in sub-elite soccer populations also apply to professional players. This study is evidence that it is possible to gather hydration related data on elite players without disruption to normal training or competitive schedules. Furthermore, the results of the present study suggest sweat [Na+] do not change over time or between different sessions. Therefore, this sweat [Na+] analysis, often the expensive and technically time-consuming part of the process, may be completed on a single occasion. The sweat [Na+] can then be applied to sweat rates, which are more easily measured on multiple occasions. However, further studies are required to confirm this applies over the entirety of the season.

Reviewer 3 Report

Dear authors,

Thank you for your paper. It is well laid out and easy to read. Sometimes I think that the language could be a little simpler to illustrate your point. Just a couple of minor issues with this for me, I have indicated this in the paper.

 I made one comment about a citation of Buono et al.

Stats - correlations or regressions?

Overall I am very positive and have made a few minor suggestions that I think will improve the paper. Perhaps my comment on limitations is not minor, please consider this. Perhaps one limitation is the different times of year of testing. I am not familiar with the schedule of La Liga, but the players may not have the same level of fitness or sweating mechanisms in summer as compared to winter. Top athletes, no question, but still perhaps an influence on the data.

I have included my comments on the attached PDF.

Thank you and congratulations on your work!

Author Response

Reviewer 3

General comment:

Dear authors,

Thank you for your paper. It is well laid out and easy to read. Sometimes I think that the language could be a little simpler to illustrate your point. Just a couple of minor issues with this for me, I have indicated this in the paper.

 I made one comment about a citation of Buono et al.

Stats - correlations or regressions?

Overall I am very positive and have made a few minor suggestions that I think will improve the paper. Perhaps my comment on limitations is not minor, please consider this. Perhaps one limitation is the different times of year of testing. I am not familiar with the schedule of La Liga, but the players may not have the same level of fitness or sweating mechanisms in summer as compared to winter. Top athletes, no question, but still perhaps an influence on the data.

I have included my comments on the attached PDF.

Thank you and congratulations on your work!

General response

Thank you for the review. Re the stats we have tried to add clarity in the statistics section re the correlation rather than regressions and made corresponding inclusions (noted by other reviews) to the results section. We have addressed the comments made throughout the pdf, and listed below corresponding to the page/line number. We appreciate the constructive comments and believe the responses have made improved the manuscript considerably. All changes to the manuscript are marked in blue text. 

Specific comment and response:

Comment:

Commented [M21]: Please carefully check the accuracy of names and affiliations. Changes will not be possible after proofreading.

Responses: Thank you. We have checked with all co-authors who have confirmed their affiliations.

Comments: Page 2:

Response: Thank you. Sentence has been corrected.

Fluid losses vary greatly… “deleted word “in””

Response: players has been changed to “players’” as suggested.

Response: Thank you. The capital letter of intake “I” has been changed to lower case.

Response: the words “have been” are added to final sentence as suggested.

…and technologies have been developed to rapidly quantify Sweat [Na+] [19]. 

Comment Page 3:

Response:

Detail has been added as to when the VO2max assessments were completed:

(V̇̇O2max assessments were performed in July).

Comment page 4:

Response: Thank you. Detail has been added to the statistic section with regard to the correlations performed and corresponding results added to the figures (also noted by other reviewers).

Comments page 9: Discussion, second paragraph last sentence.

Response: Thank you. The final paragraph has been rewritten to more specific and improve the comprehension. We agree, “simplicity is key”… though one of the most difficult of writing tasks. The sentence has been rewritten from:

The increased layers of clothing will increase sweat rates [26]. Therefore, it appears in cool conditions players fail to modify their clothing as training progresses and consistent with previous studies drink less in comparison to exercise in the heat [10].

To

Hypohydration ensues by multiple layers of clothing increasing sweat rates [26], combined with reduced fluid intake when exercising in cool environments [10].

Comment: include a sentence to underscore the importance of hydration

Response: We agree, however, we feel we adequately address the importance through the sentence:

“Thus, care should be taken that players are not simply accustomed to hypohydration.”

Followed by guidelines. To further build out practical implications are included in the additional section of “practical implications” request by review 2. Included in blue text.

Comments page 10:

Comment: Restructure sentence:

To achieve this, appropriate rehydration strategies are recommended between exercise sessions [17] and multiple measures (daily body mass, thirst, Urine USG) are used to track changes in players hydration status [31].

Response: sentence has been restructured to:

To achieve this, appropriate rehydration strategies are recommended between exercise sessions [17]. In addition, daily changes in body mass, thirst, and urine USG may be collected to track changes in players hydration status [31].

Comment: a section on the study limitations has been added, as follows:

The present study was descriptive and thus several limitations are acknowledged. First, the final assessment was completed 5 months after the first tests completed in July. It is unknown if the players acclimation and/or physical status changed over this duration, which may have influenced both the sweat rate and sweat [Na+] response to exercise. Second, the analysis of sweat did not include potassium concentration which would have been preferential as a quality control for sweat samples [43]. Despite this, best practice of sweat collection (clean skin, avoided patch saturation, gloves, etc.) described by Baker [43] were followed, so that we have confidence in our analysis.  Other factors reported to influence sweat [Na+] such as hormones [44] and diet [45] were not recorded or controlled for prior to testing, due to the nature of field/descriptive studies. Finally, as some (n=3) players ingested no carbohydrate during exercise, future studies could investigate how alternative sources of carbohydrate (i.e. gels, bars, chews) may influence ad-libitum carbohydrate intake of players. This would be relevant during high intensity training sessions, prolonged pre-season training and to encourage practicing match day nutrition.

Comment: Check referring style.

Response:

Thank you the reference list has been updated, to be consistent throughout and with Journal referencing requirements.

END.

Round 2

Reviewer 1 Report

I accept the publication of this article in this version.

Author Response

The authors would like to thank the reviewer for all comments and contributions to improving the manuscript. 

Reviewer 2 Report

The authors have made a great effort to improve the manuscript.

Author Response

(The authors gave the same response as above.)
